# Analysis of Functional Neuroplastic Changes in the Cortical Language System in Relation to Different Growth Patterns of Glioblastoma

**DOI:** 10.3390/brainsci13060867

**Published:** 2023-05-27

**Authors:** Katharina Hense, Daniel Deuter, Mark W. Greenlee, Christina Wendl, Nils Ole Schmidt, Christian Stroszczynski, Christian Doenitz, Christian Ott, Katharina Rosengarth

**Affiliations:** 1Department of Neurosurgery, University Hospital Regensburg, 93053 Regensburg, Germany; daniel.deuter@ukr.de (D.D.); nils-ole.schmidt@ukr.de (N.O.S.); christian.doenitz@ukr.de (C.D.); christian.ott@ukr.de (C.O.); katharina.rosengarth@ukr.de (K.R.); 2Institute for Experimental Psychology, University of Regensburg, 93053 Regensburg, Germany; mark.greenlee@psychologie.uni-regensburg.de; 3Department of Radiology, University Hospital Regensburg, 93053 Regensburg, Germany; christina.wendl@klinik.uni-regensburg.de (C.W.); christian.stroszczynski@klinik.uni-regensburg.de (C.S.)

**Keywords:** glioblastoma, heterogeneity, brain tumor, functional magnetic resonance imaging, blood oxygenation level dependent imaging, task-based functional connectivity, preoperative mapping, preoperative noninvasive imaging

## Abstract

The interpretation of fMRI data in glioblastoma (GB) is challenging as these tumors exhibit specific hemodynamic processes which, together with malignancy, tumor volume and proximity to eloquent cortex areas, may lead to misinterpretations of fMRI signals. The aim of this study was to investigate if different radiologically defined GB tumor growth patterns may also influence the fMRI signal, activation pattern and functional connectivity differently. Sixty-four patients with left-hemispheric glioblastoma were included and stratified according to their radiologically defined tumor growth pattern into groups with a uniform (U-TGP) or diffuse tumor growth pattern (D-TGP). Task-based fMRI data were analyzed using SPM12 with the marsbar, LI and CONN toolboxes. The percent signal change and the laterality index were analyzed, as well as functional connectivity between 23 selected ROIs. Comparisons of both patient groups showed only minor non-significant differences, indicating that the tumor growth pattern is not a relevant influencing factor for fMRI signal. In addition to these results, signal reductions were found in areas that were not affected by the tumor underlining that a GB is not a localized but rather a systemic disease affecting the entire brain.

## 1. Introduction

In addition to structural imaging, functional magnetic resonance imaging (fMRI) has been established in clinical routine for the identification and characterization of eloquent areas for different cognitive functions. This allows for the visualization of cortical activations, which are estimated using the blood-oxygenation-level-dependent (BOLD) effect [1] and included in the further surgical planning. This approach has been shown to reduce the risk of intraoperative damage to eloquent areas and thus has a positive effect on outcome [2].

Despite the many advantages this method offers to both the patient and the treating physician, its limitations and restrictions should not be ignored. For example, it has been noted in the past that different tumor-associated factors affect the fMRI signal and thus the integrity of preoperative functional imaging. In this context, it has already been shown that the presence of a tumor significantly attenuates the BOLD signal [3,4] as well as the extent of cortical activation [5]. It is assumed that the tumor affects the brain’s ability to autoregulate cerebral blood flow as a result of neuronal activity and thus also affects the hemodynamic response captured by fMRI [6].

It has also been demonstrated that the malignancy of a tumor is an important factor [7,8,9] that can have a significant effect on the level of the BOLD signal. It is assumed that especially high-grade tumors such as glioblastoma show impaired neurovascular coupling or even uncoupling, which can affect the fMRI signal and lead to misinterpretation of false-negative cortical activation. However, the influence of the tumor on the BOLD response is not exclusively limited to the area within its radiological border; it can also be found in surrounding normally vascularized areas. In particular, high-grade tumors infiltrating the surrounding tissue and altering their cell structure may affect signal intensity [5,10,11].

The consideration of tumor volume, on the other hand, has proven to be a controversial factor in previous studies. While it is assumed that with increasing volume, functional tissue is compressed and progressively limited in function, this has been shown in some studies to be a factor that has only a moderate effect [12]. Similar results were found for the influence of patient age [7,12].

In addition, the proximity of the tumor to eloquent cortex areas also plays an important role, resulting in reductions in the BOLD signal or activation patterns, more so in the tumor-affected hemisphere than the unaffected hemisphere [7,13]. In addition, patients with glioblastoma showed that the fMRI signal decreased with increasing proximity to the tumor [9]. Some authors even claim that fMRI signals in the vicinity of a tumor may even be significantly decreased due to abnormal neovascularization of a malignant brain tumor [14,15].

Glioblastoma (GB) represents an undifferentiated and exclusively malignant tumor of the highest CNS WHO grade 4. Histologically, this brain-derived tumor belongs to the category of astrocytic tumors [16,17]. The heterogenous nature of this type of tumor is described as a noteworthy hallmark [18].

In addition to its heterogeneity at the cellular and molecular level [19,20,21], the morphologically heterogeneous appearance of the tumor in magnetic resonance imaging (MRI) has also been shown to be prognostically relevant in previous studies. Both qualitative studies involving VASARI features [22] and quantitative studies using radiomics features [23] showed that these tumor manifestations could be related to survival. It has also been demonstrated that three prognostically relevant subtypes of glioblastoma can be distinguished using cluster analysis. Characteristics such as shape, texture and marginal sharpness were found to be relevant here [24]. The results described in this article have been confirmed by further studies. It was shown that a correlation between the shape of the tumor and survival can be found independently of such factors as age, overall condition or tumor volume [25].

The aim of this study was to investigate whether the tumor growth pattern in glioblastoma, in addition to the influencing factors described above, is also a parameter that should be taken into account when acquiring and evaluating preoperative fMRI results in order to provide the patient with the maximum benefit. In this study, we evaluated the differences between two patient groups with different glioblastoma tumor growth patterns with regard to BOLD signal, cortical activation patterns and functional connectivity using task-based fMRI. Since classification into different tumor growth patterns has already been shown to be prognostically relevant in previous studies, it was assumed in this study that this also affects the impact of glioblastoma on fMRI results. It was hypothesized that the prognostically more favorable group of patients with a uniform tumor growth pattern would show higher values in percent signal change as well as the laterality index and stronger connections in functional connectivity compared to the prognostically less favorable group of patients with a diffuse tumor growth pattern.

## 2. Materials and Methods

### 2.1. Study Sample

The patient cohort included language fMRI data for 64 patients with histologically confirmed glioblastoma in the left hemisphere (see Table 1), which were acquired during routine preoperative fMRI examinations. In order to incorporate different functions of the complex language system, such as phonological, semantic and syntactic processing [26], the patients completed up to three different tasks to identify language-critical brain areas, which included tasks to generate verbs, antonyms and sentences. The patients were subdivided based on the morphological appearance of the tumor in the T1-weighted MRI image after contrast agent administration (Figure 1) into groups of patients with a uniform contrast margin not thicker than 4 mm (uniform tumor growth pattern; U-TGP) or a diffuse contrast pattern with a thickness of >4 mm (diffuse tumor growth pattern, D-TGP). This subdivision was decided by an experienced neuroradiologist with more than 10 years of experience (author C.W.). The two groups did not differ statistically significantly with regard to the patients’ age.

In addition, retrospectively selected data from 32 healthy control subjects (mean age 27.82 years) who completed the same language paradigms were included in this study. These were acquired as part of a study that provided the basis for establishing fMRI protocols for patients at the University Hospital. The size of the sample was based on the number of available data.

### 2.2. Image Acquitision

Data were collected using two different MRI scanners at the University Hospital Regensburg (patient sample) and at the District Hospital Regensburg (control sample). Patients’ data were acquired using a Siemens Skyra 3-Tesla full-body scanner (MAGNETOM Skyra; Siemens, Erlangen, Germany) with a 32-channel head coil. The visual stimuli were presented via a mirror mounted on the head coil, which directed the view to an MR-compatible 32” BOLD screen (Cambridge Research Systems, Rochester, UK) placed at the end of the scanner. Functional images were acquired using a T2*-weighted gradient echo planar imaging sequence (TR = 2000 ms, TE = 30 ms, FoV = 192 × 192 mm^2^, flip angle = 90°, voxel size = 2 × 2 × 3 mm^3^, 31 slices). In addition, a T1-weighted structural image (TR = 1980 ms, TE = 3.67 ms, FoV = 256 × 256 mm^2^, flip angle = 9°, voxel size = 1 × 1 × 1 mm^3^) was obtained.

Data from healthy control subjects were acquired at the District Hospital Regensburg using a Siemens Allegra 3T head scanner (MAGNETOM Allegra; Siemens, Erlangen, Germany) with a 1-channel phased-array head coil. Visual stimuli for each task were projected onto a projection screen at the end of the scanner so the subjects could observe them via a mirror on the head coil. Functional images were acquired using a T2*-weighted EPI sequence (TR = 2000 ms, TE = 30 ms, FoV = 192 × 192 mm^2^, flip angle = 90°, voxel size = 3 × 3 × 3 mm^3^, 34 slices) as well as a high-resolution T1-weighted structural image (TR = 2300 ms, TE = 2.91 ms, FoV = 256 × 256 mm^2^, flip angle = 9°, voxel size = 1 × 1 × 1 mm^3^, 160 slices).

Stimuli were presented in written form and were centrally arranged and presented in black font on a light-gray background using the stimulus delivery and experimental control program Neurobehavioral Systems Presentation (www.neurobs.com, accessed on 21 September 2022). For this study, three language paradigms were used. During the stimulation periods, the patients and controls were instructed to generate either a semantically associated verb corresponding to a presented noun (verb generation task), the antonym of a presented adjective (antonym generation task) or a sentence from four presented words (syntax generation task). Each task was performed in a separate run, each using the same design. During the acquisition, there were alternating blocks of stimulation (20 s) and rest periods (fixation of a cross; 10 s). During the stimulation period, a new stimulus was presented every 2 s, to which the patients were asked to respond subvocally according to each particular task. This prevented movement artifacts as well as sensorimotor language-supportive but not language-critical brain activation. In this study, each of the three language paradigms used during the preoperative examination of patients was included in the analysis to control whether the results could be replicated in order to exclude paradigm-specific effects.

### 2.3. Data Analysis

The functional MRI data for each of the three language paradigms were analyzed using Statistical Parametric Mapping 12 (SPM12). Preprocessing involved realignment to correct for motion-related artifacts, coregistration of the functional and structural images and normalization of the structural and functional data into a standard MNI space. Subsequently, the functional images were smoothed with a kernel of FWHM 8 × 8 × 8 mm³ to increase the signal-to-noise ratio. Based on the general linear model, the individual design matrices were composed of a regressor that reflected each examination condition and was convolved with the hemodynamic response function. The fixation periods were not explicitly modeled as a separate regressor and served as an implicit baseline. The motion correction parameters calculated during realignment were also included as six additional regressors to reduce intraindividual variance. The corresponding language paradigm was chosen as a contrast to calculate the respective *t*-statistics.

To define the regions of interest (ROIs), anatomical masks of six language-relevant regions (angular gyrus, supramarginal gyrus, middle and superior temporal gyrus, inferior frontal gyrus opercular and triangular part) and two control regions (left and right occipital lobe) were extracted from the AAL atlas using the WFU Pickatlas Toolbox Version 3.0.5 [27,28,29]. Due to the visual component of the paradigms, the occipital lobes of both hemispheres were set as control regions since these areas were not affected by the tumor in any patient and at the same time were reliably stimulated by the chosen paradigms. These anatomical masks were used to analyze the percent signal change (PSC) using the Marsbar Toolbox Version 0.44 [30]. Within each anatomical mask, the voxel with the maximum activation based on the t-map previously generated in the statistical analysis was identified. The coordinates of this voxel were subsequently used in the Marsbar Toolbox to generate a spherical ROI with a diameter of 5 mm around this voxel, which was then used for the analysis of the PSC. ROIs with an overlap with the tumor (contrast enhancement and/or necrosis) were excluded from the analysis. This included a total of 63 ROIs, which were distributed into 29 ROIs in 15 patients during verb generation, 15 ROIs in 13 patients during antonym generation and 19 ROIs in 13 patients during syntax generation.

In addition to the percent signal change, the laterality index (LI) for each language paradigm was calculated in the areas of the frontal, parietal and temporal lobe using the LI toolbox [31] to quantify the extent of symmetry of cortical activation.

To analyze the functional connectivity, the CONN toolbox implemented in SPM12 [32] was used. As the data had already been preprocessed during data analysis in SPM12, the resulting SPM.mat files were reused. The BOLD time series were bandpass-filtered (0.008–0.09 Hz) during denoising, and the realignment parameters as well as their first-order temporal derivatives were included. Furthermore, linear detrending was performed [33] and group ROI-to-ROI analyses were computed using 23 selected regions of interest of the Default Mode network, Salience network, Dorsal Attention network, Frontoparietal network and Language network.

Numerical data such as percent signal change and laterality indices were analyzed using SPSS version 28 (IBM, Armonk, NY, USA). The respective standard errors of the mean are represented by the error bars. Statistically significant results are indicated with * for *p* < 0.05, ** for *p* < 0.01 and *** for *p* < 0.001.

## 3. Results

### 3.1. Percent Signal Change

In order to investigate the influence of the GB tumor growth pattern on the BOLD signal, the three language paradigms were evaluated and percent signal change was calculated in regions of interest (ROIs) relevant to these functions. For each of the 24 ROIs (8 ROIs per language paradigm), a one-way ANOVA was calculated. ROIs containing tumor masses were excluded for each individual patient, which summed up to a total of 63 excluded ROIs (29 ROIs in 15 patients for verb generation, 15 ROIs in 13 patients for antonym generation and 19 ROIs in 13 patients for syntax generation). The individual tests were adjusted using false discovery rate (FDR) correction for multiple comparisons.

#### 3.1.1. Verb Generation

The analysis of the verb generation paradigm (Figure 2) showed statistically significant between-group differences in the angular gyrus (F(2,87) = 5.583; p_adj_. = 0.025), inferior frontal gyrus opercular part (F(2,90) = 3.922; p_adj_. = 0.048) and triangular part (F(2,90) = 3.879; p_adj_. = 0.048), as well as the left occipital lobe (F(2,93) = 8.641; p_adj_. = 0.002). Furthermore, a trend was found in the right occipital lobe (F(2,93) = 3.042; p_adj_. = 0.090), but no statistically significant differences were identified in the middle temporal gyrus (F(2,88) = 2.334; p_adj_. = 0.154), superior temporal gyrus (F(2,88) = 2.187; p_adj_. = 0.167) or supramarginal gyrus (F(2,86) = 0.677; p_adj_. = 0.533). To further characterize these differences, post hoc t-tests (FDR-corrected) were computed. These showed statistically significant differences between patients with a diffuse tumor growth pattern (D-TGP) and healthy control subjects in the angular gyrus (p_adj_. = 0.011) and the inferior frontal gyrus opercular part (p_adj_. = 0.032). In the control region of the occipital lobe of the affected left hemisphere, control subjects differed from both patients with a D-TGP (p_adj_. = 0.006) and patients with a uniform tumor growth pattern (U-TGP, p_adj_. = 0.006). In the right unaffected occipital lobe, there was a trend of a difference between controls and D-TGP patients (p_adj_. = 0.062). A statistically significant difference between the two patient groups was not found in any of the regions of interest.

#### 3.1.2. Antonym Generation

For the antonym generation paradigm (Figure 3), we also found a statistically significant group difference in the areas of the angular gyrus (F(2,71) = 5.146; p_adj_. = 0.033), the left occipital lobe (F(2,76) = 10.279; p_adj_. = 0.001) and the superior temporal gyrus (F(2,74) = 4.408; p_adj_. = 0.040). In addition, there was a trend in the middle temporal gyrus (F(2,72) = 2.986; p_adj_. = 0.090), but not in the inferior frontal gyrus opercular part (F(2,76) = 2.057; p_adj_. = 0.180) or triangular part (F(2,75) = 1.210; p_adj_. = 0.365), nor in the right occipital lobe (F(2,76) = 2.057; p_adj_. = 0.180). Post hoc *t*-tests showed that the D-TGP group differed significantly from the control group in the angular gyrus (p_adj_. = 0.016). In addition, differences were found in the left affected occipital lobe between the control subjects and the D-TGP group (p_adj_. = 0.006) and the U-TGP group (p_adj_. = 0.005). The two patient groups differed statistically significantly in the superior temporal gyrus (p_adj_. = 0.032).

#### 3.1.3. Syntax Generation

In the syntax generation paradigm (Figure 4), a one-factor ANOVA (FDR-corrected) revealed statistically significant group differences in the areas of the angular gyrus (F(2,82) = 4.669; p_adj_. = 0.035), middle temporal gyrus (F(2,82) = 4.713; p_adj_. = 0.035) and occipital lobe of the affected left hemisphere (F(2,86) = 9.133; p_adj_. = 0.002) as well as the occipital lobe of the unaffected right hemisphere (F(2,86) = 11.331; p_adj_. = 0.001). Furthermore, there were significant differences in the inferior frontal gyrus triangular part (F(2,84) = 4.308; p_adj_. = 0.040) and a trend in the opercular part (F(2,85) = 2.980; p_adj_. = 0.090). No statistically significant differences were found in the superior temporal gyrus (F(2,83) = 0.512; p_adj_. = 0.601) or supramarginal gyrus (F(2,81) = 1.396; p_adj_. = 0.320). FDR-corrected post hoc t-tests showed a statistically significant difference between control subjects and D-TGP patients in the angular gyrus (p_adj_. = 0.006) and in the inferior frontal gyrus triangular part (p_adj_. = 0.032). Furthermore, differences were found in the occipital lobe of both the affected and unaffected hemispheres between control subjects and U-TGP patients (p_adj_. = 0.006 in each case) and between control subjects and D-TGP patients (p_adj_. = 0.006 and p_adj_. = 0.003).

Additional analyses were performed to evaluate the influence of patient sex or tumor location on the percent signal change. After adjustment for multiple testing using FDR correction, there were no statistically significant differences between the two patient groups (Figure A1 and Figure A2, Figure A3, Figure A4 and Figure A5).

### 3.2. Laterality Index

To investigate the hemispheric distribution of cortical activations, the laterality index (LI) was calculated for the three paradigms in the frontal, parietal and temporal lobes using the LI toolbox (Figure 5).

A Kruskal–Wallis test (FDR-corrected) showed a statistically significant difference in LIs in the frontal lobe (χ^2^ = 10.161; p_adj_. = 0.014) and in the parietal lobe (χ^2^ = 11.421; p_adj_. = 0.010), but not in the temporal lobe (χ^2^ = 0.027; p_adj_. = 0.987) during the verb generation paradigm. Post hoc tests showed significant differences in the frontal lobe between the control group and both the U-TGP group (p_adj_. = 0.014) and the D-TGP group (p_adj_. = 0.024). A similar result was found when comparing the parietal lobe, where both patient groups also differed from the control group (p_adj_. = 0.007 and p_adj_. = 0.017).

For the antonym generation paradigm, statistically significant differences were found in the frontal lobe (χ^2^ = 23.044; p_adj_. < 0.001) and parietal lobe (χ^2^ = 20.244; p_adj_. < 0.001) but not in the temporal lobe (χ^2^ = 4.477; p_adj_. = 0.160). Post hoc tests showed that in the frontal as well as the parietal lobes, the control group differed from both patient groups (p_adj_. < 0.001 each).

Evaluation of the LIs for the syntax generation paradigm revealed no statistically significant differences the frontal lobe (χ^2^ = 4.506; p_adj_. = 0.160), parietal lobe (χ^2^ = 2.771; p_adj_. = 0.322) and temporal lobe (χ^2^ = 1.637; p_adj_. = 0.469).

### 3.3. Functional Connectivity

In addition to percent signal change and laterality indices of language-relevant regions of interest, the functional connectivity of relevant cortical networks was analyzed. In this study, the Default Mode, Salience, Dorsal Attention, Frontoparietal and Language networks, with a total of 23 regions of interest (ROIs) implemented in the CONN toolbox, were analyzed to evaluate intra- as well as inter-network connectivity (Figure 6). In this analysis, the two patient groups, one with a uniform tumor growth pattern (U-TGP) and the other with a diffuse tumor growth pattern (D-TGP), were compared with each other as well as a healthy control group.

#### 3.3.1. Verb Generation

When comparing the D-TGP patients with the matched control subjects, we found a significantly decreased intra-network connectivity within all networks. At the same time, these patients exhibited extensive inter-network connectivity between the ROIs of the different networks, which was significantly more pronounced than in the assigned control subjects. This pattern was also evident when comparing the U-TGP patients with their matched control subjects, although fewer connections were involved. When comparing the two patient samples, the intra-network connections of the U-TGP patients within the Salience and Dorsal Attention networks as well as connections between the ROIs of the Salience, Dorsal Attention and Frontoparietal networks were significantly stronger. In addition, a significant reduction in the inter-network connectivity of several ROIs of the Default Mode network was found in this group.

#### 3.3.2. Antonym Generation

During antonym generation, both patient groups showed significantly reduced intra-network connectivity within all networks compared to their respective assigned control subjects. In addition, both patient groups also exhibited extensive inter-network connectivity between the ROIs of the different networks, this being statistically significantly stronger than in the assigned control subjects. When comparing the two patient groups, there were some significantly more pronounced connections in the D-TGP group, which were spread across all networks examined. At the same time, only the connection between the right posterior superior temporal gyrus and the right rostral prefrontal cortex was significantly reduced in this comparison.

#### 3.3.3. Syntax Generation

In the analysis of the syntax generation paradigm, the respective controls showed significantly stronger intra-network connectivity within all networks compared with their respective assigned patients. When comparing the U-TGP patients and control subjects, we found significant reductions in inter-network connectivity between the Salience and Language networks. Furthermore, there were extended inter-network connections between the ROIs of the different networks in both patient groups that were statistically significantly stronger than in the assigned control subjects. Comparison of the two patient groups also revealed statistically significant differences within and between networks, which were distributed across all networks examined.

## 4. Discussion

The aim of this study was to investigate if different glioblastoma tumor growth patterns have different influences on the BOLD signal, cortical activation patterns and functional connectivity. For this purpose, the percent signal change, laterality indices and functional connectivity of language-relevant regions of interest were analyzed and compared between two patient groups with different tumor growth patterns as well as a healthy control group. We hypothesized that the prognostically more favorable group of patients with uniform tumor growth patterns (U-TGPs) would show higher values in the percent signal change and laterality index as well as stronger intra- and inter-network connectivity compared to the prognostically less favorable group of patients with diffuse tumor growth patterns (D-TGPs).

To analyze the influence of the tumor growth pattern on the BOLD signal, the percent signal change (PSC) in language-relevant regions of interest (ROIs) was evaluated. In this study, we found a statistically significant difference between the two patient groups only in the superior temporal gyrus during antonym generation, with the patients with uniform tumor growth patterns showing a higher PSC value. Considering the other results of the percent signal change analysis, the U-TGP patients showed qualitatively higher PSC values than the D-TGP group in most regions of interest, although these were not significant at the statistical level. At the same time, however, it was noticeable that when comparing the healthy control subjects with the two patient groups, most existing differences were found in the group of patients with diffuse tumor growth patterns as compared to the group of U-TGP patients. Previous studies reported that glioblastoma with a uniform rim-enhancing tumor growth pattern exhibits lower tumor cell density, a medium edema and high sphericity [34]. Additionally, these tumors are associated with low angiogenesis and microvascularity compared to irregular and solid GB subtypes. This heterogeneity of neurovascular processes accompanying a GB is also likely to account for a variable extent of reduction in the BOLD signal, although this was not significant in this study.

In addition to the functional changes in language-relevant regions of interest, we also found reduced PSC values in the control ROIs in the occipital lobes of both patient groups compared to the healthy controls. In this study, it was noticeable that this not only applied to the affected hemisphere, but, in the case of the syntax generation task, also the contralateral unaffected hemisphere. Regarding the effect of a glioblastoma on the unaffected hemisphere, there are conflicting findings from studies in the literature. While most studies reported a reduction in BOLD signal or cortical activation of the ipsilateral hemisphere compared to the contralateral hemisphere [3,7,35], studies focused on analyzing the connectome showed the influence of the tumor on the contralateral hemisphere [36,37]. In addition, neuropathological studies also found evidence for invasion of tumor cells into healthy brain tissue and thus the systemic nature of this disease [38]. Concerning this debate, our results suggest that the tumor does not only locally affect the brain and its function, but also exerts global effects on the entire brain, including the unaffected hemisphere.

We also analyzed differences in laterality indices between the patients and control subjects and between groups of patients with uniform and diffuse tumor growth patterns. In this study, statistically significant differences were found in the verb and antonym generation paradigms between the healthy controls and each of the patient groups, but not during syntax generation. It was also noticeable that comparisons between the two patient groups did not show any statistically significant differences. This is in agreement with the differences in the percent signal change between the control subjects and both patient groups found in this study.

The evaluation of functional connectivity especially showed alterations in intra-network connectivity, which was significantly reduced in glioblastoma patients. In this study, these alterations were not only found in the affected hemisphere, but also in interhemispheric connections. Our results are in line with previous studies reporting global changes in functional connectivity accompanying a brain tumor [39,40,41]. In this context, tumors in language-critical areas of the dominant hemisphere have also been shown to induce attenuation of functional connectivity in both the affected and unaffected hemispheres during the execution of a language task within the language network [37] as well as in the Default Mode network [42]. In this study, comparison between the two groups of tumor patients with uniform and diffuse tumor growth patterns showed only marginal differences, as was already found for percent signal change and laterality indices. Therefore, it can be concluded that all three comparisons indicate only qualitative, not quantitative, differences between the groups.

### Limitations of the Study

Regarding the methodology of the study, some limitations should be noted. First, it should be mentioned that the data for the study samples, the patients and the control groups, were acquired using different MRI scanners. Although the same paradigms were used for both groups and only the location of the data collection differed, it is uncertain what influence this has on the results. This influence is assumed to be weaker for the laterality indices as well as the functional connectivity, as in these analyses either normalized quotients or, respectively, temporal correlations, were calculated. Although the influence was estimated differently in the individual comparisons due to the methodology used, it has been shown that the results of the respective analyses agree with each other and are mutually consistent.

It should also be noted that the subjects of the control group were notably younger than the patients. The influence of age-related changes in brain function can therefore not be completely excluded in comparing the patient group with the healthy control subjects. However, the influences of the different MRI scanner, examination protocol and age can all be considered minor when comparing the two patient groups, since the data of these two groups, whose comparison was the focus of this study, were collected in the same clinical setting and, moreover, the groups did not differ statistically significantly with respect to age.

Moreover, it should be noted that the groups of patients differed in terms of tumor volume in the verb generation paradigm, although the size of edema did not show a statistically significant difference. This may have affected the fMRI signal. However, on the basis of the data, we hypothesize that this would further minimize the overall minor differences between the two groups and therefore would not quantitatively alter the results described above. This is further supported by the analyses of the antonym and syntax generation paradigms, which showed similar results, although in these analyses there was no statistically significant difference between the two patient groups.

While most studies dealing with language function and fMRI include only right-handed subjects, this was not possible in our study due to its retrospective nature. Although the data for the healthy control group were acquired under controlled conditions, where only right-handed subjects were included, this information was not available for the two patient groups, as this hardly has any relevance in the context of treatment. Since handedness may affect the lateralization of language in the brain, we controlled for this in the patients and found two patients with right lateralized language function. The exclusion of these two patients, however, did not change the results of the study.

In addition, the professions of the subjects and patients were not considered in this study, as no individual information was available on this in either group. However, it was known that the control group consisted mainly of students. We attempted to minimize possible differences between groups resulting from job-related language utilization by using a large sample size.

## 5. Conclusions

In conclusion, direct comparison of the two groups of patients with uniform and diffuse tumor growth patterns showed only minor and mainly statistically non-significant differences. Considering these results, we assume that the radiological tumor growth pattern, which was distinguished based on the morphological appearance of contrast enhancement, is not a relevant factor influencing the fMRI signal. Moreover, when control regions distant from the tumor were examined, it was shown that the influence of the tumor on the BOLD signal is not only locally limited to the area of the glioblastoma, but also has global effects on functional activity of the contralateral unaffected hemisphere.

## Figures and Tables

**Figure 1 brainsci-13-00867-f001:**
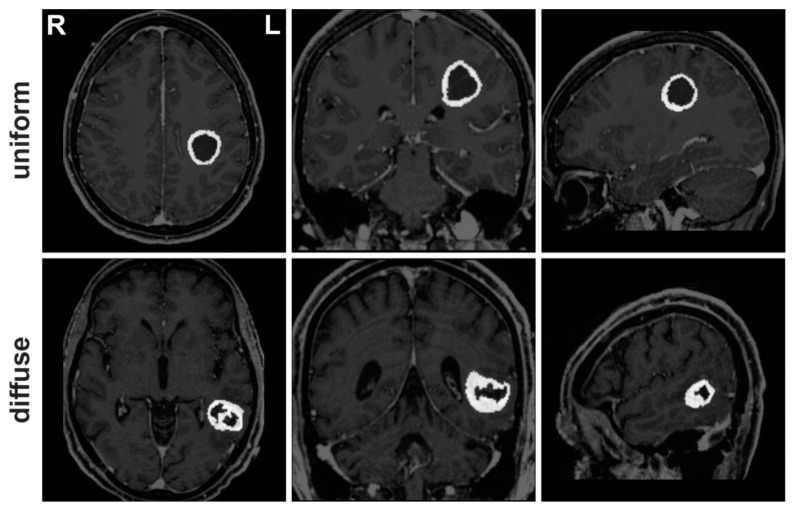
Exemplary contrast-enhanced T1-weighted MRI images of patients with a uniform tumor growth pattern (upper row) and a diffuse tumor growth pattern (lower row). For better visualization, the contrast enhancement is highlighted in the MR image. Abbreviations: L left; R right.

**Figure 2 brainsci-13-00867-f002:**
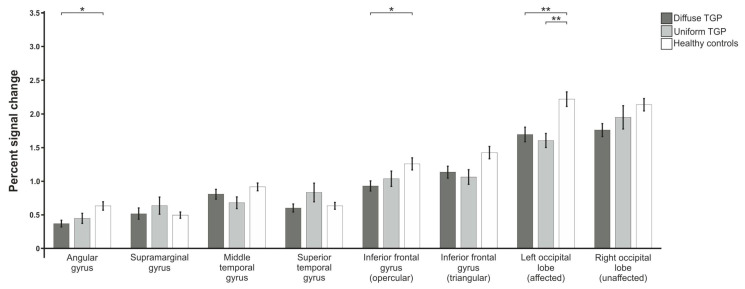
Comparison of mean percent signal changes of the three groups of patients with uniform and diffuse tumor growth pattern as well as healthy control subjects for the verb generation paradigm. Statistically significant results are indicated with * for *p* < 0.05, ** for *p* < 0.01.

**Figure 3 brainsci-13-00867-f003:**
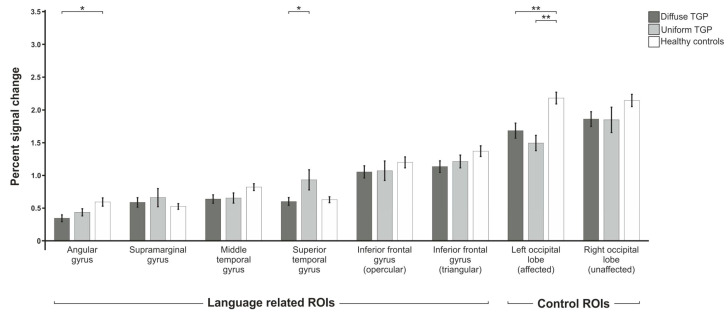
Comparison of mean percent signal changes of the three groups of patients with uniform and diffuse tumor growth patterns as well as healthy control subjects for the antonym generation paradigm. Statistically significant results are indicated with * for *p* < 0.05, ** for *p* < 0.01.

**Figure 4 brainsci-13-00867-f004:**
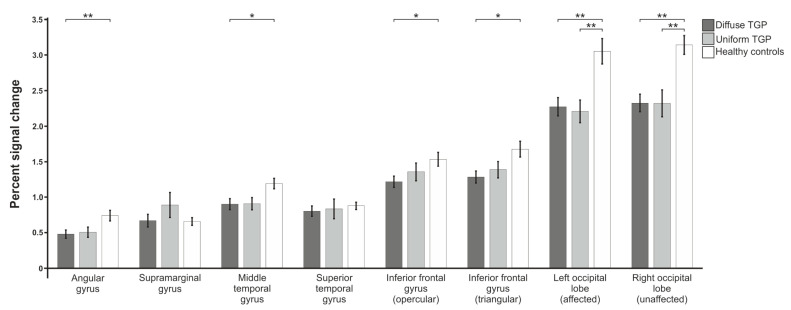
Comparison of mean percent signal changes of the three groups of patients with uniform and diffuse tumor growth pattern as well as healthy control subjects for the syntax generation paradigm. Statistically significant results are indicated with * for *p* < 0.05, ** for *p* < 0.01.

**Figure 5 brainsci-13-00867-f005:**
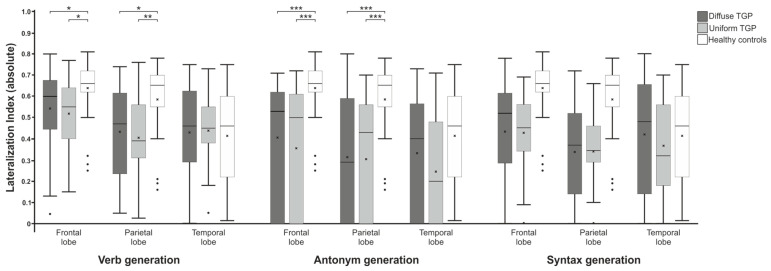
Comparison of the laterality indices of the language paradigms in the fontal, parietal and temporal lobes between patient groups with uniform and diffuse tumor growth patterns and the control group. Statistically significant results are indicated with * for *p* < 0.05, ** for *p* < 0.01 and *** for *p* < 0.001.

**Figure 6 brainsci-13-00867-f006:**
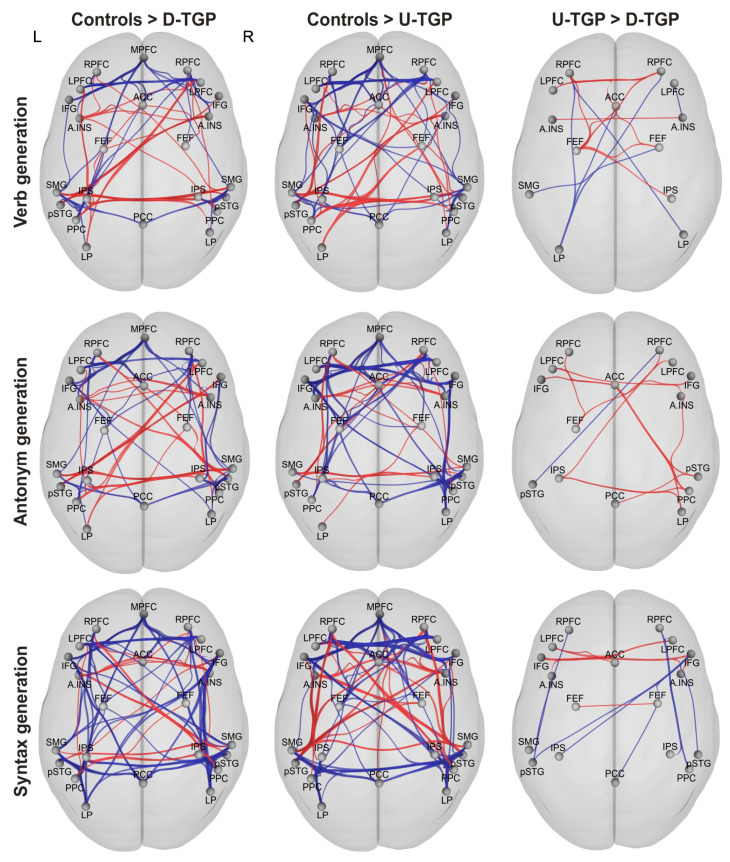
Comparisons of functional connectivity of cortical networks relevant for the processing of language between the control group and each patient group as well as between both patient groups with uniform and diffuse tumor growth patterns. Here, red signifies connections that were more pronounced in the U-TGP group while blue connections were more pronounced in the D-TGP group, with thicker lines implying stronger connections. Abbreviations: L left; R right.

**Table 1 brainsci-13-00867-t001:** Demographic data of the patient sample as well as the subsamples of patients with uniform and diffuse tumor growth patterns. Depending on tumor location and general condition, patients completed up to three different fMRI language tasks, which included the generation of verbs, antonyms and sentences. Statistically significant results are indicated with * for *p* < 0.05.

		Total	Uniform Tumor Growth Pattern	Diffuse Tumor Growth Pattern	Sig.
Verb generation	*n*	64	23	41	
	Age	59.84	62.87	58.15	0.114
	Sex (m/f)	37/27	13/10	24/17	
	Tumor location (frontal/parietal/temporal)		5/4/14	13/9/19	
	Tumor size in mm^3^		22,707.65	35,876.75	0.031 *
	Edema size in mm^3^		52,534.35	56,229.12	0.707
Antonym generation	*n*	47	16	31	
	Age	59.66	62.31	58.29	0.274
	Sex (m/f)	26/21	9/7	17/14	
	Tumor location (frontal/parietal/temporal)		2/4/10	9/8/14	
	Tumor size in mm^3^		24,437.44	36,302.55	0.098
	Edema size in mm^3^		58,471.71	57,401.19	0.926
Syntax generation	*n*	57	22	35	
	Age	59.86	63.90	57.83	0.095
	Sex (m/f)	33/24	13/9	20/15	
	Tumor location (frontal/parietal/temporal)		5/4/13	8/9/18	
	Tumor size in mm^3^		22,536.14	34,112.34	0.076
	Edema size in mm^3^		52,936.82	54,296.62	0.894

## Data Availability

The data presented in this study are available on request from the corresponding author. The data are not publicly available due to patients’ privacy.

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
