# Peer review of "Analysis of Functional Neuroplastic Changes in the Cortical Language System in Relation to Different Growth Patterns of Glioblastoma"

_brainsci, 2023, doi:10.3390/brainsci13060867_

Round 1
Reviewer 1 Report
The authors performed an interesting experiment on brain function in patients with glioblastoma. However, the work has some shortcomings. - The controls are of different ages than the patients. What can affect the obtained result. Also, authors should write how they chose people as controls. - The authors analyzed the BOLD signal while performing language tests. For this reason, control persons should be selected on the basis of the same profession as patients. A linguist should not be compared to a manual worker in this test. - The authors should only include right-handed patients in their studies. - Authors should also compare the results by patient gender.
Reviewer 2 Report
Manuscript number: brainsci-2393394
Title: Analysis of functional neuroplastic changes in the cortical language system in relation to different growth patterns of glioblastoma
Summary
The authors investigated how different patterns of tumor growth in glioblastoma (GB) affect the interpretation of functional magnetic resonance imaging (fMRI) data. Sixty-four patients with left-hemispheric GB were included and stratified into two groups based on radiologically defined tumor growth patterns: uniform (U-TGP) or diffuse (D-TGP). The results showed non-significant differences in fMRI data between the two patient groups, indicating that tumor growth pattern does not have a significant influence on the fMRI signal. However, the authors found signal reductions in areas of the brain that were not affected by the tumor, suggesting that GB is a systemic disease affecting the entire brain, rather than a localized one.
While this article is educational, it is unsuitable for publication for the reasons set out below.
1. In Materials and Methods, the authors mentioned “The patient cohort included language fMRI data of 64 patients with histologically confirmed glioblastoma in the left hemisphere (see Table 1)”. However, Fig. 1 showed GB in the right hemisphere.
2. A more detailed description of the location of the tumor is needed to objectively prove the analysis results.
3. The authors classified them into two groups according to the contrast enhancement pattern at the margin of the tumor. In order to distinguish the two groups more objectively, it seems necessary to describe various characteristics such as tumor size and peritumoral edema.
Thank you very much.
Round 2
Reviewer 1 Report
The authors answered all the reviewer's doubts.
Authors should write a chapter about the limitations of their research based on their responses to the review.
Reviewer 2 Report
Dear authors
Thank you very much for your responses.
I’ve read the responses for reviewer’s comments.
Revisions are generally satisfactory.
I feel that this manuscript contains valuable information worthy of publication in Brain Sciences.
Thank you.
